# Genome-Wide Association Analysis of Freezing Tolerance and Winter Hardiness in Winter Wheat of Nordic Origin

**DOI:** 10.3390/plants12234014

**Published:** 2023-11-29

**Authors:** Gabija Vaitkevičiūtė, Aakash Chawade, Morten Lillemo, Žilvinas Liatukas, Andrius Aleliūnas, Rita Armonienė

**Affiliations:** 1Institute of Agriculture, Lithuanian Research Centre for Agriculture and Forestry, 58343 Akademija, Lithuania; zilvinas.liatukas@lammc.lt (Ž.L.); andrius.aleliunas@lammc.lt (A.A.); rita.armoniene@lammc.lt (R.A.); 2Department of Plant Breeding, Swedish University of Agricultural Sciences, 230 53 Alnarp, Sweden; aakash.chawade@slu.se; 3Faculty of Biosciences, Institute of Plant Science, Norwegian University of Life Sciences (NMBU), 1432 Ås, Norway; morten.lillemo@nmbu.no

**Keywords:** climate change, cold acclimation, GWAS, overwintering, *Triticum aestivum* L.

## Abstract

Climate change and global food security efforts are driving the need for adaptable crops in higher latitude temperate regions. To achieve this, traits linked with winter hardiness must be introduced in winter-type crops. Here, we evaluated the freezing tolerance (FT) of a panel of 160 winter wheat genotypes of Nordic origin under controlled conditions and compared the data with the winter hardiness of 74 of these genotypes from a total of five field trials at two locations in Norway. Germplasm with high FT was identified, and significant differences in FT were detected based on country of origin, release years, and culton type. FT measurements under controlled conditions significantly correlated with overwintering survival scores in the field (*r* ≤ 0.61) and were shown to be a reliable complementary high-throughput method for FT evaluation. Genome-wide association studies (GWAS) revealed five single nucleotide polymorphism (SNP) markers associated with FT under controlled conditions mapped to chromosomes 2A, 2B, 5A, 5B, and 7A. Field trials yielded 11 significant SNP markers located within or near genes, mapped to chromosomes 2B, 3B, 4A, 5B, 6B, and 7D. Candidate genes identified in this study can be introduced into the breeding programs of winter wheat in the Nordic region.

## 1. Introduction

The temperate zone is favorable for the cultivation of winter wheat (*Triticum aestivum* L.), which produces higher yields compared to spring wheat due to the longer growing period [1]. As a result of global climate change, the increasing growing season temperatures in temperate regions are predicted to lead to higher grain yields, especially if spring wheat cultivation is switched to winter wheat [2]. However, as more favorable climate conditions for the cultivation of winter wheat are shifting to higher latitudes, winter hardiness will remain an essential trait to be included in breeding programs. The prolonged and warmer autumns in conjunction with more frequent temperature fluctuations will negatively affect the hardening and the subsequent winter survival of winter wheat [3,4]. Moreover, the stagnating yields of European wheat cultivation [5] provide additional challenges for winter wheat breeding programs. Consequently, winter-hardy and freezing-tolerant cultivars will be needed to meet the growing demand for food.

Winter hardiness or overwintering in the field is a complex trait, consisting of multiple factors, such as tolerance to desiccation, ice-encasement, or snow mold [6,7,8]. However, freezing tolerance (FT) is the primary component of winter hardiness [9]. FT is achieved through cold hardening, also known as cold acclimation, which occurs in winter wheat throughout a period of low above-freezing temperatures in autumn. This process induces a cascade of morphological and molecular changes. Depending on the genotype, cold acclimation lasts up to 8 weeks [10,11]. However, significant differences in FT among cultivars can be observed even after 2 weeks [12]. The major regulators of FT in winter wheat are known as the *Frost resistance 1* (*Fr-1*) and *Fr-2* loci on chromosome 5A [13]. *Fr-1* is linked with the vernalization locus *Vrn1*, which regulates the dormancy and initiation of the reproductive phase in response to temperature [14]. Moreover, the *Fr-A2* locus contains a group of at least 18 *C-REPEAT BINDING FACTOR* (*CBF*) genes, which are upregulated by exposure to low temperatures [15]. The copy number variation and polymorphism of these genes have been shown to directly influence winter hardiness in wheat [16,17]. At present, studies have linked multiple quantitative trait loci (QTLs) on chromosomes 1A, 1B, 1D, 2A, 2B, 2D, 3A, 3D, 4A, 5A, 5B, 6D, and 7D of winter wheat to FT [18,19,20]. The candidate genes, discovered in such studies, can be used in breeding programs for winter wheat with enhanced FT.

The panel of winter wheat of Nordic origin, provided by the Nordic Genetic Resource Centre (NordGen, Alnarp, Sweden) offers a rich repository of genetic diversity, which could be applied in future breeding efforts of new climate-resilient crops. This panel consists of cultivars and landraces, which in this study will be collectively referred to as “cultons”, a term for cultivated plants proposed by Hetterscheid and Brandenburg [21]. These cultivars and landraces had previously been utilized to investigate resistance to Septoria tritici blotch [22], powdery mildew [23], and drought [24]. Morphological and agronomic traits have likewise been assessed in some genotypes from this collection [25]. However, to our knowledge, FT has not yet been evaluated in this diverse panel of genotypes. Winter wheat, originating from Northern European countries, is likely to contain unique alleles, contributing to the strong adaptation to their local environment. Notably, wheat landraces have previously been proposed as a good source of genetic diversity for cultivar improvement [26,27]. Therefore, the evaluation of FT and subsequent genome-wide association studies (GWAS) in this diverse panel of Nordic cultivars and landraces would provide valuable information about candidate genes associated with FT. GWAS is an effective biometrics-based method, allowing researchers to associate genetic variations within a population with specific observed traits [28]. Such studies have been used to associate markers with yield, disease resistance [22,29], grain quality [30], and abiotic stress tolerance [15,18,31]. The aims of this study were (1) to assess the FT of 160 winter wheat cultivars and landraces of Nordic origin; (2) to determine the relationship between FT under controlled conditions and overwintering in the field over three winter seasons and two locations; and (3) using GWAS to identify the markers and candidate genes associated with FT and winter hardiness.

## 2. Results

### 2.1. Freezing Tolerance of Winter Wheat under Controlled Conditions

To evaluate the FT of 160 Nordic genotypes, freezing tests under controlled conditions were carried out. A strong positive correlation (*r* = 0.76) was determined between the FT, assessed throughout two freezing experiments, carried out under controlled conditions in 2021 and 2022 (*p* < 0.0001) (Appendix A). FT was evaluated as LT_50_ mean values from these two experiments, which followed a normal distribution (Figure 1A) and were used for the subsequent analyses. The broad-sense heritability of FT under controlled conditions was estimated at 0.87. The genotypes were classified according to country of origin, year of release, and culton type (Figure 1B–D).

The LT_50_ values of the Nordic winter wheat genotype collection after 2 weeks of cold acclimation at 2 °C ranged between −10.57 and −14.34 °C, observed in the least freezing-tolerant genotype ‘Solist’ and the most freezing-tolerant genotype ‘Vakka’, respectively. The most freezing-susceptible genotypes originated from Denmark, whereas the most freezing-tolerant genotypes were of Finnish origin (*p* < 0.0001) (Figure 1B; Appendix A). Swedish and Norwegian genotypes displayed intermediate levels of FT. One genotype of German origin and two genotypes of unknown origin were removed from this analysis due to the small sample size (Appendix A). Genotypes released in 1941–1980 were more freezing-tolerant in comparison to those released in 1896–1940 and 1981–2012 (*p* < 0.0001) (Figure 1C; Appendix A). The release year of 23 genotypes was unknown; however, these genotypes exhibited the highest levels of FT compared to the remaining groups (*p* < 0.0001). Landraces showed higher freezing tolerance in comparison with cultivars (*p* < 0.001) by an average of 0.76 °C (Figure 1D; Appendix A). The genotypes were thus classified into three FT groups according to their LT_50_ results: less tolerant, intermediate, and more tolerant (Appendix A).

The genotypic data, consisting of 7401 single nucleotide polymorphisms (SNPs), were applied to perform PCAs and visualize and compare the separation of 160 genotypes across countries of origin, years of release, culton types, and level of FT under controlled conditions (Figure 2). Dimension 1 and dimension 2 accounted for 11.8% and 9.6% of variation, respectively. Despite the higher number of SNP markers used, the clustering patterns here were similar to those reported by Odilbekov et al. [22], where Denmark and Finland formed two distinct groups (Figure 2A). Dimension 2 separated the genotypes released between 1896 and 1980 from genotypes released in 1981 and later (Figure 2B). The PCA plots reveal that the majority of the landraces fall into the more freezing-tolerant cluster (Figure 2C,D; Appendix A). Additional data regarding minor allele frequencies (MAF) are provided in Appendix A. Furthermore, the kinship matrices of 160 winter wheat cultivars and landraces, tested under controlled conditions, and 74 genotypes, tested in the field, are available in Appendix A, respectively.

### 2.2. Overwintering of Winter Wheat in the Field

Field trials were conducted to determine the winter hardiness of the NordGen winter wheat collection. The overwintering of 74 winter wheat genotypes from the same panel was assessed in five separate field experiments, consisting of two locations and three winter seasons: Eidum 2008–2009, 2009–2010, and 2010–2011, and Vollebekk 2008–2009 and 2009–2010. The overwintering survival ranged from 0 to 66.7% in Eidum 2008–2009; between 0 and 100% in Eidum 2009–2010; 0–90.6% in Eidum 2010–2011; from 0 to 100% in Vollebekk 2008–2009; and between 13.9 and 100% in Vollebekk 2009–2010. The broad-sense heritability of winter hardiness in the field was estimated to be 0.53.

Correlations were assessed between the mean LT_50_ values of 74 winter wheat genotypes and overwintering survival percentages in each of the five field trials. Significant (*p* < 0.05) correlations were found between LT_50_ data and Eidum 2008–2009 (*r* = −0.45), Eidum 2009–2010 (*r* = −0.25), Eidum 2010–2011 (*r* = −0.61), and Vollebekk 2008–2009 (*r* = −0.41) overwintering values (Appendix A). The correlations were negative due to higher LT_50_ values indicating lower FT. Thus, a highly significant relationship was identified between FT under controlled conditions and overwintering in the field, with Eidum 2010–2011 showing the strongest correlation (Figure 3). The least freezing-tolerant genotypes, such as the Danish ‘Lading Skæghvede’ or ‘Ideal’, and the most freezing-tolerant genotypes, such as the Finnish ‘Jyvä’ or the Swedish ‘Pärl II’, showed the same FT patterns under both controlled and field conditions.

### 2.3. GWAS of Freezing Tolerance under Controlled Conditions

To find significant associations between FT under controlled conditions and SNP markers, GWAS analyses were carried out. Six significant SNP markers were associated with FT under controlled conditions using BLINK and FarmCPU models (Figure 4). These markers were filtered by assessing the effect of their alleles on LT_50_ of winter wheat. The alleles of five markers had a significant effect (*p* < 0.05) on FT under controlled conditions (Figure 5; Table 1).

Four of these markers, found on chromosomes 2A, 2B, 5A, and 5B, were detected by both models. One marker, located in chromosome 7A, was identified solely by FarmCPU. The marker BobWhite_c23903_443 was located within 3799 bp of an uncharacterized gene, while BobWhite_c28133_87 and Excalibur_c2598_2052 were located within uncharacterized genes, with the former marker yielding an amino acid substitution (p.A374V) (Appendix A). The marker Kukri_c14902_1112 on chromosome 2B was found to lie within a putative *1-phosphatidylinositol-3-phosphate 5-kinase FAB1C* gene and lead to an amino acid substitution (p.I934V). Moreover, the marker RAC875_c16644_491 in chromosome 7A was located in an intron of the *FAR1-Related sequence 5-like* gene (Appendix A).

### 2.4. GWAS of Overwintering in the Field

GWAS analyses were conducted to detect significant associations between overwintering in the field and SNP markers. Three separate GWAS analyses were carried out for Eidum 2008–2009, Eidum 2010–2011, and Vollebekk 2008–2009 overwintering field trials. The 2008–2009 Eidum trial yielded 14 significant SNP markers on chromosomes 2B, 2D, 3B, 4A, 5B, 6B, 7B, and unknown chromosomes (Figure 6). Following the assessment of marker allele effects on overwintering scores, eight significant markers were retained (Table 2; Appendix A). The marker Excalibur_c43822_370 on chromosome 3B was detected by five different models (BLINK, FarmCPU, GLM, MLM, and MLMM). It was found to be within 3252 bp of *TraesCS3B01G412500*, which codes for an ethylene-responsive transcription factor (TF), ABR1-like (Appendix A). The marker wsnp_Ex_c607_1204908, identified by BLINK, FarmCPU, and MLMM, was located on chromosome 5B within 89 bp of *TraesCS5B01G010700*—a gene encoding a CSC1-like protein, RXW8.

Two markers were aligned to unknown chromosomes; however, the sequence analyses revealed their underlying genes and locations. The marker Tdurum_contig47476_528 was detected by GLM and MLM and was situated in an uncharacterized gene, mapped to chromosome 4A. The alleles of this marker result in a silent mutation (p.D122D) (Appendix A). Kukri_rep_c85536_598, likewise identified by GLM and MLM, was located within a *lupeol synthase-like* gene on chromosome 7D, where it results in another silent mutation (p.S566S).

The four remaining markers (Kukri_c57491_156, RFL_Contig3621_1157, Tdurum_contig47476_495, and RAC875_c88279_291) were associated with genes coding for an uncharacterized LOC123045374 (p.L141L), a peroxisomal acyl-coenzyme A oxidase 1-like protein (p.L358L), an uncharacterized LOC123088501 (p.E111E), and a putative F-box protein At2g02030 (p.R390R), respectively (Appendix A).

One significant SNP marker was identified in the Eidum 2010–2011 trial—Tdurum_contig50731_961 in chromosome 5B (Table 2; Appendix A). This marker is located in an intron of a gene encoding an rRNA-processing protein, EFG1-like (Appendix A).

Two significant SNP markers were associated with the Vollebekk 2008–2009 trial overwintering data (Table 2; Appendix A). Marker wsnp_JD_c10233_10936535 on chromosome 3B was identified by the models BLINK and MLMM, whereas the marker BobWhite_c18566_106 on chromosome 6B was detected by FarmCPU. The marker wsnp_JD_c10233_10936535 is located in a gene coding for a transcription initiation factor, TFIID subunit 7-like protein, which results in a silent mutation (p.A26A). Marker BobWhite_c18566_106 was located in an uncharacterized gene yielding an amino acid substitution (p.N2K) (Appendix A).

## 3. Discussion

### 3.1. The Geographical and Temporal Trends of Freezing Tolerance in Winter Wheat

The cultivation of winter-type crops is shifting to higher latitudes due to increasingly favorable climate conditions during the growing season [2]. However, the low negative temperatures in winter can still cause yield loss in the form of winterkill. Consequently, it is important that crops cultivated in Nordic countries possess a high level of winter hardiness. Although winter hardiness is a complex trait, FT is its major component [9]. Our results show that the level of FT of the 160 tested winter wheat genotypes reflected the latitude of their geographical origins, with Danish genotypes exhibiting the lowest levels of FT, and Finnish genotypes being the most freezing-tolerant, whereas Swedish and Norwegian genotypes had intermediate FT (Figure 1B). Winter wheat is cultivated at latitudes ranging from 46° N to 61.34° N [32,33], and earlier studies have reported similar patterns, with Northern European winter wheat genotypes exhibiting higher levels of FT, and Southern genotypes having less FT [34]. FT is a complex multi-genic trait achieved through cold acclimation, and thus, numerous environmental factors can affect the winter survival of crops [4,35]. Due to the strong natural and artificial selection processes, the cultivars and landraces are usually well adapted to the local environmental conditions, such as photoperiod, irradiance, soil and air temperature in winter, winter length, and spring temperature. For example, a specific combination of the vernalization (*Vrn*), photoperiod (*Ppd*), and earliness *per se* (*Eps*) alleles determines whether winter wheat is more suitable for cultivation under longer and colder winters in Northern Europe, or shorter and milder winters in Southern Europe [34]. Similarly, Sthapit Kandel et al. [27] reported a correlation between winter wheat genotypes’ latitude of origin and their FT. Thus, winter wheat cultivars originating from Finland are highly adapted to low-temperature survival in winter, whereas selection under the milder temperate climate of Denmark results in less freezing-tolerant genotypes.

The majority of cultivated wheat had consisted of landraces up until the 19th century, and wheat breeding efforts had only truly begun with the approach of the 20th century [36]. Although the subsequent selection of modern wheat had taken into account resistance to diseases or abiotic stresses, traits associated with quality and quantity of yield had remained at the forefront of breeding programs [37,38]. Moreover, the Green Revolution in the 1960s and 1970s [39,40], while crucial to global food security efforts, was also an extremely significant factor contributing to the narrowing of the genetic pool of crops [41,42]. In our study, high levels of FT were observed in the genotypes released in between 1941 and 1980. In comparison, winter wheat released before 1941 and after 1980 showed a similar, lower range of FT (Figure 2C). According to EEA and Twardosz and Kossowska-Cezak [43,44], the winters in Europe were colder than the baseline temperature by at least 1 °C in 1887–1892, 1928, 1938–1942, and 1962–1963. The baseline temperature in these estimates was determined using the period of 1880–1899. Notably, the winter of 1941–1942 is known as the coldest European winter of the 20th century [45]. However, no winters with average temperatures below the baseline by 1 °C had been recorded after 1963, and the average winter temperatures continued to increase. This higher occurrence of cold winters in the early and middle 20th century may have influenced the selection process of winter-type crops to focus on increased winter hardiness. Moreover, while both world wars resulted in decreased agricultural output, the post-war period was marked by advances in agrotechnology, which led to the recovery and continual increase in yield production in Europe [46,47,48]. Thus, renewed crop-breeding programs had provided new freezing-tolerant cultivars, which were in higher demand after the exceptionally cold winters. This demand likely declined with gradually warming winters, which led to decreased selective pressure and resulted in wheat cultivars being less freezing-tolerant from 1981 and onwards.

Notably, the highest FT was observed in the landraces, which as a group were more freezing-tolerant than cultivars (Figure 1D and Figure 2C,D). Landraces are varieties of crops grown in a specific area over an extended period of time, and thus, both through natural selection and artificial breeding, landraces are strongly adapted to the local environment [42]. Wheat landraces tend to have earlier heading and maturation dates, more seed shattering, higher likelihood of lodging, lower spike density and are usually taller in comparison with cultivars [49]. Cavanagh et al. and Lopes et al. [50,51] propose that wheat landraces can be a good source of genetic diversity for improved and climate-resilient crops. This is likewise shown by Dotlačil et al. [26], who found that their selected winter wheat landraces had higher crude protein content and could be used to improve winter hardiness. A study by Sthapit Kandel et al. [27] also investigated a collection of winter wheat landraces and reported significant markers related to FT. Therefore, these freezing-tolerant landraces obtained from Nordic countries can be a valuable source of genetic diversity for new, improved cultivars. However, the integration of FT-related genes in new cultivars can be difficult, as under mild winter conditions, genotypes with enhanced FT can produce lower yields in comparison to the less freezing-tolerant genotypes [52,53]. Therefore, to optimize yields, future wheat genotypes should be climate-adaptable and genetically suited to their geographical locations of cultivation.

### 3.2. SNPs Associated with Freezing Tolerance under Controlled Conditions

Although the wheat genome had already been sequenced and published in 2018 [54], there is still a lack of information regarding the function of wheat genes and their protein products, as many sequences in the databases are annotated using gene prediction models and thus, not tested in vivo [55]. Moreover, the hexaploid and repetitive nature of the wheat genome provides its own challenges, with multiple copies of the same gene, called homoeologs, scattered across the three subgenomes A, B, and D [54]. Further complications arise from ancient translocation events, which occasionally result in homologous genes residing on non-homoeologous chromosomes [56]. Additionally, transposable elements (TEs) can change or disrupt coding sequences [57,58]. Such and similar factors can result in duplicates or inaccurate mapping of markers and transcripts to the reference genome. Notably, the published wheat genome originated from the spring wheat cultivar ‘Chinese Spring’, whereas the object of our study is European winter wheat, which is genetically differentiated [59]. Consequently, in our analyses, the marker BobWhite_c28133_87 was mapped to the gene *TraesCS2A01G159600LC* in chromosome 2A; however, the highest identity and query cover in a BLAST search led to the uncharacterized *LOC123180258*, mapped to chromosome 1D (Appendix A). A single CD was found in the translated protein sequence of this gene, described as a MuDR family transposase. These regions are thought to serve as transposases for mutator transposable elements, which play a role in genetic and epigenetic variation [60]; however, their function is not yet completely understood. The SNP results in an amino acid substitution (p.A374V) in this gene. Alanine and valine are both nonpolar amino acids; however, valine is more hydrophobic than alanine [61]. The exact effect of these amino acid substitutions should be investigated in further studies.

The marker Kukri_c14902_1112 on chromosome 2B was associated with a gene coding for a putative 1-phosphatidylinositol-3-phosphate 5-kinase FAB1C. The membranes of eukaryotes contain phosphatidylinositol (PtdIns)—a regulatory phospholipid, which plays multiple roles in membrane transport, vacuolar organization, and stomatal closure [62]. The FAB1C kinase adds a phosphate to phosphatidylinositol 3-phosphate (PtdIns3P) to create phosphatidylinositol 3,5-bisphosphate PtdIns(3,5)P_2_. *Arabidopsis thaliana* (L.) Heynh. *FAB1C* mutants display decreased rates of stomatal closure; however, the function and importance of this kinase are yet to be clarified [63]. Nevertheless, the role of *FAB1C* in stomatal closure could explain its importance in FT, as an earlier study had reported decreased stomatal conductance in barley to be associated with lower FT [64]. The SNP marker Kukri_c14902_1112 causes a conservative amino acid substitution in this gene (p.I934V). Isoleucine and valine are the most common amino acid substitutions resulting from a single nucleotide base change [65].

QTLs, located in group 5 chromosomes, had previously been shown to control the winter growth habit in wheat [14,18]. Studies on rye (*Secale cereale* L.), the most freezing-tolerant cereal crop, likewise revealed that genes found in group 5 chromosomes determine winter hardiness [66,67]. Here, we report two markers in the chromosomes 5A (Excalibur_c2598_2052) and 5B (BobWhite_c23903_443). Soleimani et al. [18] likewise found the Excalibur_c2598_2052 marker, which is located near the *Frost Resistance A2* (*FR-A2*) locus, to be significantly associated with FT, and identified multiple C-Repeat Binding Factor (CBF) genes in this QTL region. The CBF genes on the *Frost Resistance* (*Fr-2*) locus in rye likewise play a major role in FT [67]. Using a BLAST search, we determined that the marker is located within the uncharacterized gene *LOC123104451*. The conserved domain (CD) search showed *LOC123104451* to contain two plant homeodomain (PHD) finger domains, described to be zinc-ion binding and protein binding. PHD finger domains are highly conserved across species and play a role in histone post-translational modification and thus, the regulation of gene expression [68]. The marker BobWhite_c23903_443 on chromosome 5B was located in close proximity to the *Vernalization-B1* (*Vrn-B1*) locus. *Vrn-B1* determines the winter growth habit and FT in wheat [14]. Within 3.8 kb of this marker, the uncharacterized gene *TraesCS2B01G217700* was detected, and the highest similarity *T. aestivum* transcript was found to be *LOC123113134*. Unfortunately, no CDs were found within the protein product of this gene; therefore, its function is difficult to predict without additional data.

### 3.3. SNPs Associated with Overwintering in the Field

Winter hardiness or overwintering in the field is affected by multiple biotic and abiotic factors, which determine the survival of wheat throughout autumn and winter. Among these factors are low temperature, drought, waterlogging, ice-encasement, snow mold, pests, etc. [7,8]. For example, Kruse et al. [69] address this complex issue in their study by investigating the QTLs associated with FT and snow mold tolerance in winter wheat grown in the field and report a QTL on chromosome 5A, linked with the *FR-A2* locus, associated with both stresses. Furthermore, snow cover depth plays an important role in overwintering, especially towards the second half of winter, as a snow cover depth of 8–10 cm can ensure the survival of winter wheat when air temperatures drop as low as −27 °C [70]. Nevertheless, the consistent presence of snow cover throughout the winter increases the effect of such diseases as snow molds upon the survival of plants [7].

The GWAS for field overwintering was carried out using the data of 74 winter wheat cultivars and landraces, obtained from two locations over three winter seasons. The overwintering scores from the Vollebekk 2009–2010 trial did not show a significant correlation with LT_50_ values, obtained under controlled conditions; therefore, this trial was not included in the GWAS analyses. From the remaining four field trials, some significant markers were associated with the overwintering scores from three field trials: Eidum 2008–2009, Eidum 2010–2011, and Vollebekk 2008–2009. The winter of 2008–2009 was marked by a lack of snow cover from mid-December to mid-January, while the temperatures decreased to −10 °C in both Eidum and Vollebekk (Appendix A). The meteorological conditions in Eidum during the winter of 2010–2011, however, included a constant presence of snow cover from November towards the second half of January, ranging from 5 to 50 cm. Throughout this period, the mean air temperature fluctuated from 0 to −16 °C. It was followed by a period of unstable snow cover and freezing temperatures, which could be the likely cause of the winterkill of susceptible cultivars. Overall, the snow cover depth during the Eidum 2008–2009 trial was the thinnest in comparison to the remaining field trials. The lack of snow cover in the second half of December 2008 and at the beginning of February 2009 provided favorable conditions for winterkill as the temperatures dropped to −10 °C. FT values under controlled conditions have been shown to better correlate with overwintering scores in the field, particularly when there is less snow cover [34,71]. Consequently, this trial yielded the most—a total of 14—significant markers during the GWAS analysis.

Two markers were simultaneously identified by multiple GWAS models in the Eidum 2008–2009 trial. The marker Excalibur_c43822_370 on chromosome 3B was associated with *TraesCS3B01G412500*, coding for an ethylene-responsive TF, ABR1-like (Appendix A). ABR1 proteins are TFs, which downregulate abscisic acid (ABA) signaling in plants. Moreover, *ABR1* expression has been shown to increase during drought and cold stress in rice [72]. The marker wsnp_Ex_c607_1204908 on chromosome 5B was linked to *TraesCS5B01G010700*—a gene encoding a CSC1-like protein, RXW8. CSC1 is a calcium-permeable cation channel protein located in the cellular membrane; however, its function is not yet fully understood [73]. Nevertheless, calcium signaling is known to play a role in response to low-temperature stress, as cold leads to increased concentrations of calcium in the cytosol, activation of calcium-permeable channels, and upregulation of genes containing calcium-regulated promoter elements [74]. The remaining six markers in the Eidum 2008–2009 trial were detected by single models and additional discussion regarding these markers and genes can be found in Appendix B.

A single marker Tdurum_contig50731_961, identified in the Eidum 2010–2011 trial, was located in *TraesCS5B02G446000* in chromosome 5B—a gene encoding an rRNA-processing protein, EFG1-like. The EFG1 protein is involved in rRNA processing through the assembly and reorganization of 18S rRNA [75]. Although the SNP is located within the intron of this gene, certain mutations in non-coding regions have previously been shown to lead to disruption of gene translation and RNA splicing, subsequently resulting in an altered protein product [76]. Finally, two markers from the Vollebekk 2008–2009 trial data were associated with overwintering. The marker wsnp_JD_c10233_10936535 was located in *TraesCS3B01G265400* in chromosome 3B, which codes for a transcription initiation factor, TFIID subunit 7-like protein. The TF complex TFIID is composed of multiple subunits and plays a role in the recognition of promoters and the initiation of transcription by RNA polymerase II, thus, ultimately mediating the expression of genes in response to external factors [77]. The Marker BobWhite_c18566_106 was located in an uncharacterized gene *TraesCS6B01G008800LC* in chromosome 6B. A BLAST search revealed the *T. aestivum* gene *LOC123135359* as the most identical. The subsequent CD analysis showed a single REALLY INTERESTING NEW GENE (RING)-finger protein domain. Proteins containing this domain usually act as an E3 ubiquitin ligase in the ubiquitination process of proteins. These proteins play numerous roles in plant growth, development, stress resistance, and signal transduction [78]. RING-finger proteins have previously been reported to negatively affect FT in *Arabidopsis* and rice [79,80]. The SNP marker BobWhite_c18566_106 leads to an amino acid substitution (p.N2K). Asparagine and lysine are both hydrophilic amino acids. Asparagine-to-lysine substitutions have been shown to affect the conformation and binding affinity of proteins [81,82]. Nevertheless, the candidate genes discussed in this study should be examined separately to accurately assess their effect on FT and to determine whether the specific alleles found in this panel of Nordic cultivars and landraces could be introgressed into new climate-resilient varieties of winter wheat.

Extreme temperatures or temperature fluctuations during winter are becoming more prevalent due to global climate change. Winter hardiness is a complex trait influenced by multiple factors such as resistance to biotic factors, snow cover, ice encasement, and level of FT [6,7]. However, FT plays a major role in winter survival under unstable snow cover. FT is especially important under inadequate snow cover conditions, and it was observed in numerous cases when the frost spells inflicted major damage to crop species not protected by snow [34,71]. Therefore, FT represents a trait of prime importance when introducing novel germplasm into breeding programs, especially in temperate climate zones. The assessment of FT under field conditions is quite unreliable due to the unpredictable variation of agrometeorological conditions during winters and usually requires manual snow removal from the plots, which can damage the plants. Due to this, we observed no reliable differentiation for winter hardiness of the 160 Nordic winter wheat genotype collection propagated in Akademija, Lithuania (55°23′ N, 23°57′ E), during two 2018–2019 and 2019–2020 winter seasons. The overwintering data from the Eidum 2010–2011 trial showed the highest correlation with FT under controlled conditions (Figure 3). As expected, the genetic relationships between genotypes were reflected by their levels of FT; however, the fluctuations in winter hardiness were notably more prominent. For example, Extra Squarehead, Iduna, and Bore, which were all descended from Squarehead, had similar LT_50_ values, differing by 0.24 °C, whereas their overwintering survival score ranged from 13.9 to 56.9%. The absence of common genetic markers found between FT under controlled conditions and field trials indicates the discrepancy in environmental conditions between the experiments. This finding is also valid between the marker–trait associations identified in different field trials in Norway due to the variations in environmental conditions throughout different seasons. Therefore, a freezing test under controlled conditions remains a reliable method to assess the levels of FT for winter wheat accessions. Moreover, FT under controlled conditions showed significant correlations with winter hardiness in the field trials, where the negative temperatures were lower with less snow cover. Phenotypic and genomic selection are promising strategies for improving FT in winter wheat. However, winter hardiness represents a much more complex trait, which is highly dependent on currently unpredictable environmental conditions. Thus, it is a challenge to obtain consistently reliable winter hardiness estimates. The strategies for improving winter hardiness in wheat could encompass the improvement of individual traits, such as FT, resistance to biotic factors, and ice encasement, using phenomic and genomic tools. The best-performing germplasm could then be combined to develop cultivars designed for future climate scenarios.

## 4. Materials and Methods

### 4.1. Plant Material and Growth Conditions

One hundred and sixty winter wheat genotypes, comprising cultivars and landraces of Nordic origin, were chosen for this study (Appendix A) [22]. This collection was obtained from NordGen. Genetic relationships between the cultivars were examined using NIAB global wheat pedigree files [83] and the Helium crop pedigree visualization software v. 1.19.09.03 [84]. The seeds were placed on filter paper in Petri dishes, soaked in water, and stored at 4 °C in the dark for 4 days. They were then transferred to room temperature for 16 h. Ten imbibed seeds of each genotype served as a single replicate and were sown into a single 125 cm^3^ well of a 28-well tray containing a peat moss substrate (Durpeta, Šepeta, Lithuania). A total of 3 replicates of every genotype were sown in a randomized pattern for every freezing test. The seedlings were grown in a greenhouse with an 18 °C temperature and a 12 h photoperiod until the three-leaf stage was reached. The wheat was transferred to a phytotron (PlantMaster, CLF Plant Climatics GmbH, Wertingen, Germany) and exposed to 2 weeks of cold acclimation at 2 °C. Cold acclimation conditions consisted of 80% relative air humidity, 200 μmol m^–2^ s^–1^ light intensity, and a 12 h photoperiod.

### 4.2. Freezing Tolerance Tests under Controlled Conditions

Freezing tests were carried out at the target temperatures of −8, −10, −12, −14, and −16 °C. These temperatures were chosen to cover the range from 0% and 100% survival, and thus, ensure the reliable assessment of LT_50_ values. This range was established in our earlier studies [34]. Moreover, two Lithuanian winter wheat genotypes (‘Ada’ and ‘Kena DS’) with known LT_50_ values were included in the freezing tests as a control. Prior to each freezing test, the trays were drenched with cold water and the plants were counted. Tests were conducted in the freezing chamber PE 2412 UY-LX (Angelantoni Industrie, Massa Martana, Italy). Thermocouple probes and a KD7 data logger (Lumel, Zielona Góra, Poland) were used to record the substrate temperature at crown depth every 2 min. The chamber temperature was gradually decreased from 2 to −6 °C over 6 h and held until the substrate temperature stabilized. Subsequently, the chamber temperature was decreased at a rate of 1 °C/h and held at each target temperature for 24 h. Following each freezing test, the temperature was raised to 4 °C at a rate of 1 °C/h and held for a further 16 h. Subsequently, the freezing chamber was switched off and the plants were kept in the dark for 18 h as the chamber temperature equalized with the room temperature. The plants were cut 2 cm above the crown region and transferred to 18 °C in the greenhouse. The numbers of regrown and dead plants were recorded after three weeks. The LT_50_ (temperature at which 50% of plants are killed) was determined using the R package “MASS” v. 7.3-60 [85]. The freezing tests were repeated twice for each target temperature.

### 4.3. Overwintering Field Experiments

A total of 5 field trials, consisting of 74 winter wheat cultivars and landraces from the same collection, were conducted in two locations in Norway over 3 winter seasons of 2008–2009, 2009–2010, and 2010–2011. The trials were sown as hillplots with 3 replicates of each cultivar using alpha-lattice designs at the “Eidum Øvre” farm, Stjørdal (63°26′ N, 10°58′ E), and the Vollebekk research farm, NMBU, Ås (59°39′ N, 10°45′ E). The 2008–2009, 2009–2010, and 2010–2011 Eidum trials were sown on the 20th, 18th, and 11th of September, respectively. The 2008–2009 and 2009–2010 Vollebekk trials were sown on the 25th and 23rd of September, respectively. Overwintering was evaluated as a visual score of the percentage of surviving plants after winter.

Daily meteorological data, including mean temperature, precipitation, and snow depth, for the Eidum trials were downloaded from the Trondheim airport Værnes weather station (63°27′ N, 10°55′ E) (Appendix A). Mean temperature and precipitation data for the Vollebekk trial were downloaded from the NMBU Ås weather station (59°39′ N, 10°46′ E) (Appendix A). Vollebekk snow depth data were not available in digital form; however, graphs from the official weather reports for Ås in 2009 and 2010 were provided (Appendix A) [86,87].

### 4.4. Genome-Wide Association Studies

The winter wheat has previously been genotyped using a 20K SNP wheat marker array, as described by Odilbekov et al. [22]. The SNP variants were aligned to IWGSCv1.0 [54], obtained from the GrainGenes database [88]. A total of 7401 markers were applied in GWAS analyses, which were carried out via the “GAPIT” package v. 3.0. This package provides a number of different GWAS methods. Here, we applied the Bayesian-information and linkage-disequilibrium iteratively nested keyway (BLINK), the fixed and random model circulating probability unification (FarmCPU), the general linear model (GLM), the mixed linear model (MLM), and the multiple loci mixed model (MLMM) [89]. The number of principal components (PCs) was set to default for the LT_50_ data, whereas the field data were analyzed using 4 PCs. The threshold for significant marker–trait associations was set to 0.05 with adjusted false discovery rate (FDR) correction applied.

Significant SNP markers were additionally filtered by testing the effect of their corresponding two most prevalent alleles on LT_50_ values and overwintering scores. Markers showing significant effects on a phenotypic trait (*p* < 0.05) were retained for further analyses. The sequences and synonyms of SNP markers were obtained from the Triticeae Toolbox (T3) repository [90]. The locations of significant SNP markers and underlying or nearby genes were determined on the GrainGenes IWGSCv1.0 Genome Browser. Nucleotide and amino acid sequences were aligned and compared using the BLAST tool [91]. The CDs of uncharacterized gene products were analyzed via NCBI’s Conserved Domain Database [92]. Amino acid changes were determined by aligning the SNP marker sequences and corresponding gene sequences on MEGA X v. 10.1.7 [93]. A MUSCLE alignment algorithm with default parameters was applied.

The variance components for broad-sense heritability estimation were calculated using the R package “lme4” v. 1.1-34 [94]. Broad-sense heritability was calculated as follows:$$H^{2}=\sigma_{G}^{2}/(\sigma_{G}^{2}+\sigma_{GE}^{2}/nEnv+\sigma_{e}^{2}/nEnv*nrep)$$
where $H^{2}$ is broad-sense heritability, $\sigma_{G}^{2}$ represents the genetic variance, $\sigma_{GE}^{2}$ is the genotype-environment interaction variance, $\sigma_{e}^{2}$ is the error variance, nEnv is the number of environments, and nrep represents the number of replications. 

### 4.5. Statistical Analyses

Statistical analyses were conducted using R v. 4.1.1 [95] and the R package “agricolae” v. 1.3-6 [96]. The Shapiro–Wilk test was applied to test the normality of data, and variances were assessed using Levene’s test. Data were analyzed via Wilcoxon Rank Sum and Kruskal–Wallis H- and Spearman’s rank correlation tests. Allele frequencies were derived from the genotypic data according to Gauch et al. [97], where common allele = 0 and rare allele = 1. These data were analyzed using principal component analyses (PCAs) from the R package ”factoextra” v. 1.0.7 [98].

## 5. Conclusions

This study shows that the NordGen gene bank collection of winter wheat genotypes contains a high range of diversity for FT and winter hardiness. The genotypes’ country of origin, year of release, and culton type had a significant effect on the FT of winter wheat. The highest FT was observed in genotypes originating from Finland, whereas the genotypes originating from Denmark were the least freezing-tolerant, with LT_50_ values ranging between −14.34 and −10.57 °C, respectively. Winter wheat released between 1941 and 1980 had significantly higher levels of FT compared to cultivars and landraces released earlier or later. Notably, the highest FT was observed in a group of 15 landraces, representing the Nordic winter wheat germplasm before the onset of systematic plant breeding. Moreover, the results of this study show that LT_50_ values under controlled conditions correlate with field overwintering scores. A total of five SNP markers were associated with FT under controlled conditions and were located in chromosomes 2A, 2B, 5A, 5B, and 7A. Additionally, 11 markers and genes were associated with FT over three field overwintering trials. These candidate genes were mapped to chromosomes 2B, 3B, 4A, 5B, 6B, and 7D. Future studies are required to validate the function of the candidate genes and to determine their involvement in FT. Thus, Nordic winter wheat germplasm can be used as a source of enhanced FT and applied in the breeding of winter wheat cultivars with improved climate resilience in the Nordic region.

## Figures and Tables

**Figure 1 plants-12-04014-f001:**
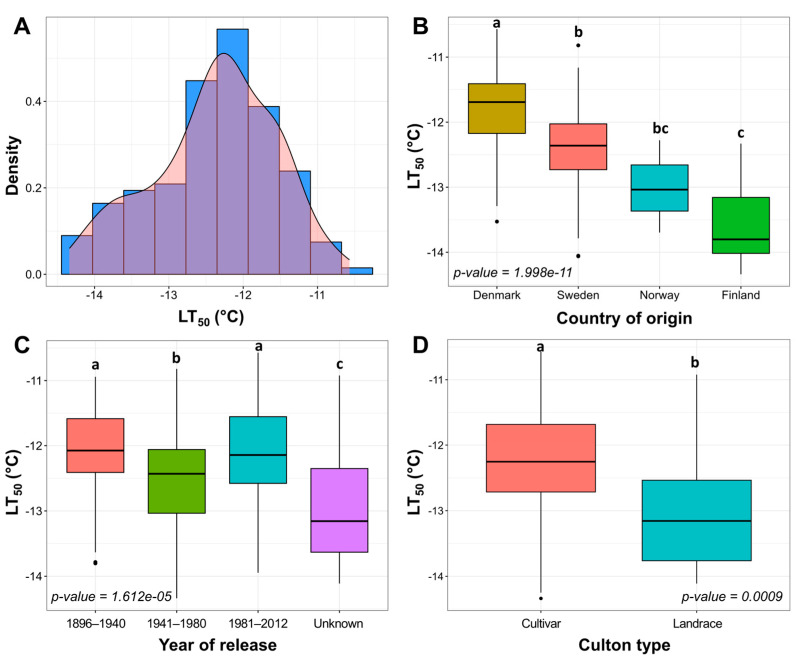
Freezing tolerance (FT) under controlled conditions in 160 cultivars and landraces of Nordic origin. Density histogram of mean LT_50_ values of two freezing tests (**A**). Variation of FT was assessed between separate groups of country of origin (**B**), year of release (**C**), and culton type (**D**). The letters above the boxplots indicate significant (*p* < 0.05) differences between compared groups.

**Figure 2 plants-12-04014-f002:**
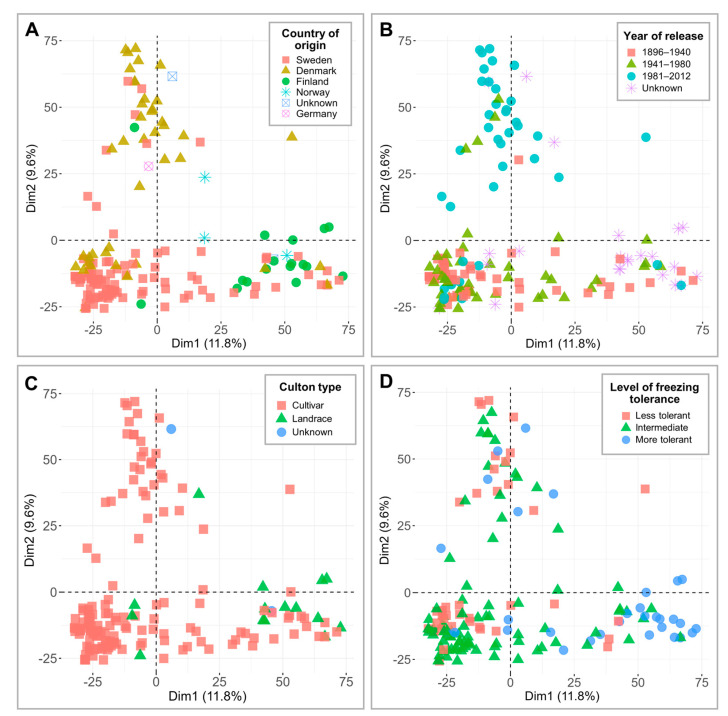
Principal component analyses (PCAs) of 7401 single nucleotide polymorphism (SNP) markers in 160 winter wheat genotypes. The results are grouped according to country of origin (**A**), year of release (**B**), culton type (**C**), and level of freezing tolerance under controlled conditions (**D**).

**Figure 3 plants-12-04014-f003:**
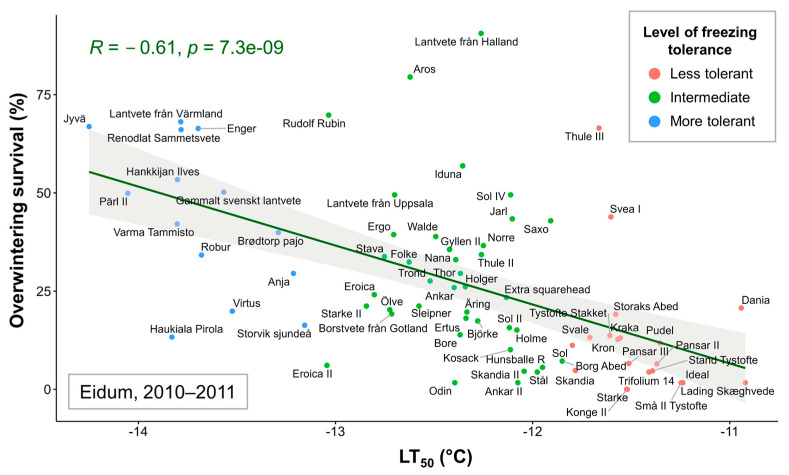
The relationship between freezing tolerance (FT) under controlled conditions (LT_50_ values, representing the temperature at which 50% of plants are killed) and FT in the field (percentage of surviving individuals after winter in the Eidum 2010–2011 field trial) of 74 winter wheat genotypes. The genotypes were grouped according to the LT_50_ values into less tolerant (down to −11.82 °C), intermediate (down to −13.08 °C), and more tolerant (down to −14.34 °C) groups.

**Figure 4 plants-12-04014-f004:**
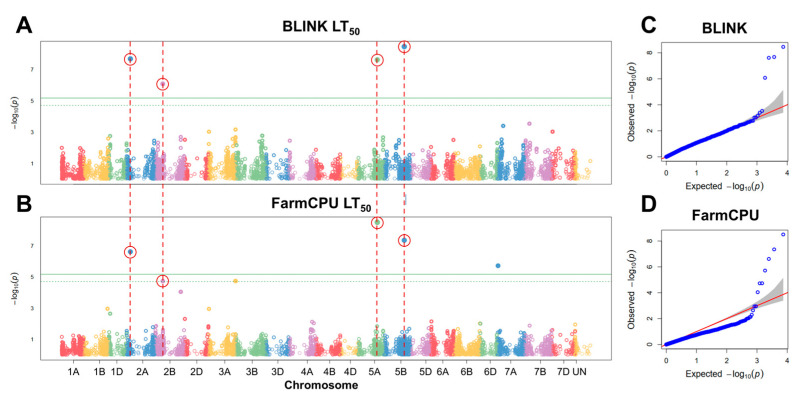
Manhattan plots of single nucleotide polymorphisms (SNPs) associated with freezing tolerance (FT), obtained using the GAPIT BLINK (**A**) and FarmCPU (**B**) models, and corresponding quantile–quantile (Q–Q) plots (**C**,**D**). FT was evaluated as LT_50_ (temperature at which 50% of plants are killed). The horizontal solid green line represents the Bonferroni cutoff, whereas the horizontal dashed green line represents the FDR cutoff. The SNPs significantly associated with FT according to both models are circled in red. The gray areas in the Q–Q plots indicate the 95% confidence interval under the null hypothesis that there is no association between the SNP and the investigated trait. Blue dots above the grey area represent the SNPs, associated with the investigated trait.

**Figure 5 plants-12-04014-f005:**
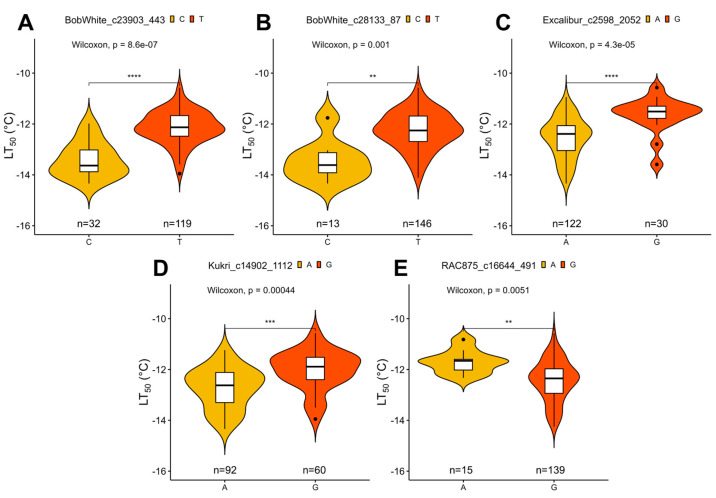
The effect of single nucleotide polymorphism (SNP) marker alleles on freezing tolerance (FT) of winter wheat under controlled conditions (LT_50_ values, representing the temperature at which 50% of plants are killed). Depicted are the markers with significant (*p* < 0.05) allele effect: BobWhite_c23903_443 (**A**), BobWhite_c28133_87 (**B**), Excalibur_c2598_2052 (**C**), Kukri_c14902_1112 (**D**), and RAC875_c16644_491 (**E**). ** indicates significant differences at *p* < 0.01, *** at *p* < 0.001, and **** at *p* < 0.0001. “n” refers to number of observations for each of the two major alleles within the population.

**Figure 6 plants-12-04014-f006:**
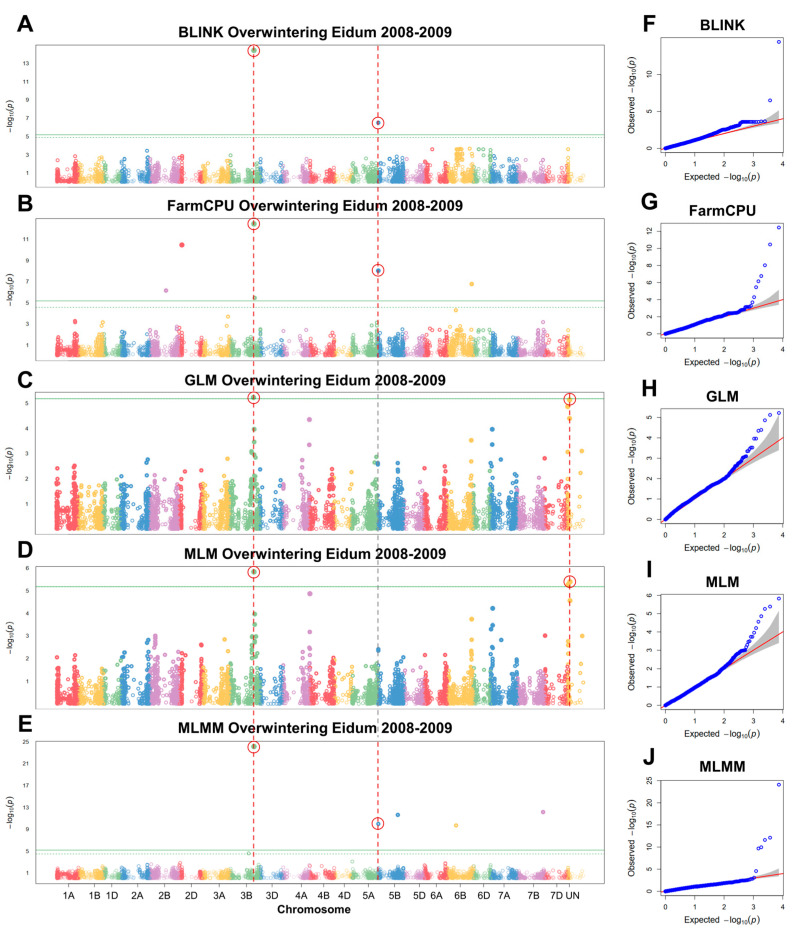
Manhattan plots of single nucleotide polymorphisms (SNPs) associated with winter survival, obtained using the GAPIT BLINK (**A**), FarmCPU (**B**), GLM (**C**), MLM (**D**), and MLMM (**E**) models, and corresponding quantile–quantile (Q–Q) plots (**F**–**J**). Winter survival was evaluated as the percentage of surviving plants after winter in the Eidum 2008–2009 field trial. The horizontal solid green line represents the Bonferroni cutoff, whereas the horizontal dashed green line represents the FDR cutoff. The SNPs significantly associated with winter survival according to multiple models are circled in red. The gray areas in the Q–Q plots indicate the 95% confidence interval under the null hypothesis that there is no association between the SNP and the investigated trait. Blue dots above the grey area represent the SNPs, associated with the investigated trait.

**Table 1 plants-12-04014-t001:** Significant SNP markers associated with freezing tolerance (LT_50_ values) under controlled conditions.

SNP Marker	Chromosome	Physical Position	MAF	Alleles	GWAS Model	Effect
BobWhite_c23903_443	5B	548427978	0.23	T/C	BLINK ****	0.33
					FarmCPU ***	0.28
BobWhite_c28133_87	2A	102022208	0.08	T/C	BLINK ****	0.44
					FarmCPU ***	0.37
Excalibur_c2598_2052	5A	519951564	0.21	A/G	BLINK ****	0.28
					FarmCPU ****	0.29
Kukri_c14902_1112	2B	206373209	0.4	A/G	BLINK **	0.19
					FarmCPU *	0.17
RAC875_c16644_491	7A	18887907	0.11	G/A	FarmCPU **	−0.33

SNP—single nucleotide polymorphism; MAF—minor allele frequency. * indicates significant differences at *p* < 0.05, ** at *p* < 0.01, *** at *p* < 0.001, and **** at *p* < 0.0001.

**Table 2 plants-12-04014-t002:** Significant SNP markers associated with overwintering in the field.

SNP Marker	Chromosome	Physical Position	MAF	Alleles	GWAS Model	Effect
**Eidum 2008–2009**						
Excalibur_c43822_370	3B	648752466	0.11	C/T	BLINK ****	17.67
					FarmCPU ****	11.68
					GLM *	14.39
					MLM *	16.48
					MLMM ****	22.07
wsnp_Ex_c607_1204908	5B	10438244	0.03	T/C	BLINK **	−12.49
					FarmCPU ****	−11.08
					MLMM ****	−13.93
Kukri_c57491_156	2B	440825074	0.22	T/C	FarmCPU ***	6.60
Tdurum_contig47476_528	UN	51199268	0.10	C/T	GLM *	14.94
					MLM*	16.36
Kukri_rep_c85536_598	UN	11380	0.11	T/C	GLM *	−13.51
					MLM *	−15.16
RFL_Contig3621_1157	4A	742335140	0.08	G/A	MLM *	−15.35
Tdurum_contig47476_495	UN	51199301	0.08	G/A	MLM *	−15.28
RAC875_c88279_291	6B	246361296	0.16	T/C	MLMM ****	−9.19
**Eidum 2010–2011**						
Tdurum_contig50731_961	5B	617710009	0.49	A/C	BLINK ***	14.54
**Vollebekk 2008–2009**						
wsnp_JD_c10233_10936535	3B	424792590	0.13	C/A	BLINK ***	−26.47
					MLMM *	−26.02
BobWhite_c18566_106	6B	3883636	0.29	C/A	FarmCPU *	−13.61

SNP—single nucleotide polymorphism; MAF—minor allele frequency. * indicates significant differences at *p* < 0.05, ** at *p* < 0.01, *** at *p* < 0.001, and **** at *p* < 0.0001.

## Data Availability

The data presented in this study are available in the Appendix A included with this paper and the paper by Odilbekov et al. [22].

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
