# Peer review of "Genome-Wide Association Analysis of Freezing Tolerance and Winter Hardiness in Winter Wheat of Nordic Origin"

_plants, 2023, doi:10.3390/plants12234014_

Round 1
Reviewer 1 Report
Comments and Suggestions for Authors
The authors evaluated the freezing tolerance (FT) of a panel of 160 winter wheat genotypes of Nordic origin under controlled conditions and compared the data with winter hardiness of 74 of these genotypes from a total of five field trials at two locations in Norway. Genome-wide association studies (GWAS) revealed 6 single nucleotide polymorphism (SNP) markers, associated with FT under controlled conditions, mapped to chromosomes 2A, 2B, 3A, 5A, 5B, and 7A. Field trials yielded 17 significant SNP markers, located within or near genes, mapped to chromosomes 2B, 2D, 3B, 4A, 5B, 6B, 7B, and 7D. However, I think this is an unfinished work, which is mainly reflected in the following aspects:
1. It is necessary to classify accessions with relevant significant SNPs, and observe phenotypic differences of accessions after classification from traits to clarify the influence of SNPs on phenotype;
2. SNPs located on genes need to be analyzed for their effects on genes;
3. The response of candidate genes to FT needs to be analyzed. Otherwise what's the point of finding these SNPs?
Comments on the Quality of English LanguageThe abstract is not vey clear; The first sentence of each section of the results should briefly explain why the experiment was designed.
Author Response
The authors would like to thank the reviewers for their constructive comments and suggestions. We have carefully considered the comments and tried our best to address every one of them. Please find included our responses to the reviewers and the descriptions of any changes made to our manuscript entitled “Genome‐Wide Association Analysis of Freezing Tolerance and Winter Hardiness in Winter Wheat of Nordic Origin”. The line numbers in our responses refer to the lines of the revised manuscript with tracked changes.
Thank you for your consideration of this manuscript.
Response to Reviewer 1
- It is necessary to classify accessions with relevant significant SNPs, and observe phenotypic differences of accessions after classification from traits to clarify the influence of SNPs on phenotype.
The markers, previously detected by GWAS, were additionally filtered by assessing the effect of their alleles on the freezing tolerance (FT)-related phenotypic traits of winter wheat. New figures (Figure 5; Figures S4,6,8), showing the effect of alleles on LT50 values and overwintering scores were added to accommodate this request. The “Results”, “Discussion”, and “Materials and Methods” sections were updated to include these new analyses.
Additionally, the SNP data of every accession are provided in Table S1. In this table, the genotypes are sorted according to LT50 values, from lowest to highest (thus, from most freezing-tolerant to least freezing-tolerant.). The overwintering data from the field trials is likewise included in this table.
- SNPs located on genes need to be analyzed for their effects on genes.
The following changes were made in accordance with this comment:
Line 563. The sentences “Significant SNP markers were additionally filtered by testing the effect of their corresponding two most prevalent alleles on LT50 values and overwintering scores. Markers showing significant effect on a phenotypic trait (p < 0.05) were retained for further analyses. The sequences and synonyms of SNP markers were obtained from the Triticeae Toolbox (T3) repository [90]” were added.
Line 571. The sentences “Amino acid changes were determined by aligning the SNP marker sequences and corresponding gene sequences on MEGA X v. 10.1.7 [93]. MUSCLE alignment algorithm with default parameters was applied.” were added.
Table S1. The table was updated to contain information about SNP alleles and the resulting amino acid changes, where applicable (columns “Nucleotide change (position in NCBI Reference sequence)”; “Amino acid change (position in NCBI Reference sequence)”). Several SNP markers were detected in the inter-gene region or introns and thus, amino acid changes could not be detected.
The “Results”, “Discussion”, and “Materials and Methods” sections were updated to include these new analyses.
- The response of candidate genes to FT needs to be analyzed. Otherwise what's the point of finding these SNPs?
The nucleotide and amino acid changes in the candidate genes are now included in Table S6 and discussed in the manuscript (details are provided in the earlier answers).
The same Nordgen panel of winter wheat had not previously been evaluated for FT, therefore, one of the main aims of our study was to assess the FT of 160 of these genotypes and landraces under controlled conditions. These results provide novel insights into the phenotypic diversity of this panel. The second aim was to determine the relationship between FT under controlled conditions and overwintering in the field over 3 winter seasons and 2 locations. Our results show that freezing tests under controlled conditions are a reliable complementary high-throughput method for FT evaluation. The third aim was to carry out GWAS analyses and identify the markers and candidate genes, associated with FT and winter hardiness in this Nordgen panel of winter wheat. We identified candidate genes, which will be further analysed in-depth in future studies.
We are planning to select the most and least freezing-tolerant genotypes from the Nordgen panel and to investigate the effects of different alleles on the phenotype (FT under controlled conditions and in the field) in our future study. The reference sequence of wheat is still rather new and contains many uncharacterized genes with unknown functions. The genes, containing the markers with identified amino acid-changing alleles, such as BobWhite_c28133_87, Kukri_c14902_1112, BobWhite_c18566_106, as well as those, containing the markers in their intronic regions (RAC875_c16644_491, Tdurum_contig50731_961) will be assessed. Finally, allelic variants found outside the genes can likewise have a significant effect on phenotypic response in eukaryotes.
- The abstract is not very clear.
To accommodate this comment, native English speakers and a qualified linguist were consulted to improve the quality of the abstract and manuscript.
- The first sentence of each section of the results should briefly explain why the experiment was designed.
The explanations were included in each section of the results in accordance with the Reviewer’s comment:
Line 83. The sentence “To evaluate the FT of 160 Nordic genotypes, freezing tests under controlled conditions were carried out.” was added.
Line 129. The sentence “Field trials were conducted to determine the winter hardiness of the NordGen winter wheat collection.” was added.
Line 155. The sentence “To find significant associations between FT under controlled conditions and SNP markers, GWAS analyses were carried out.” was added.
Line 195. The sentence “GWAS analyses were conducted to detect significant associations between overwintering in the field and SNP markers.” was added.
List of changes made, by line
(The line numbers refer to the lines of the revised manuscript with tracked changes)
Lines 11-24. The grammar and syntax of the abstract were revised, and the significant markers, as well as their locations, were updated after an additional filtering step.
Lines 13, 252, 293. “winter type” was changed to “winter-type”.
Line 62. “Hetterscheid and Brandenburg in 1995” was changed to “Hetterscheid and Brandenburg [21]”.
Lines 82, 128, 154, 194, 251, 319, 391, 499, 514, 534, 551, 581. All subsection titles were corrected in accordance with the Instructions for Authors.
Line 83. The sentence “To evaluate the FT of 160 Nordic genotypes, freezing tests under controlled conditions were carried out.” was added.
Line 115. “(2019)” was deleted.
Line 117. “Figure 2B” was referenced.
Line 119. “Figure 2 B-D” was changed to “Figure 2 C,D”. Here, we referred to Figure 2 sections “C” and “D”, the figure reference was updated accordingly.
Line 129. The sentence “Field trials were conducted to determine the winter hardiness of the NordGen winter wheat collection.” was added.
Lines 155-171. The sentence “To find significant associations between FT under controlled conditions and SNP markers, GWAS analyses were carried out.” was added. The paragraphs were updated to contain relevant information after the additional filtering of SNP markers and investigation of amino acid substitutions.
Lines 180-187 (Figure 5). new figure was added. Description was included: “Figure 5. The effect of single nucleotide polymorphism (SNP) marker alleles on freezing tolerance (FT) of winter wheat under controlled conditions (LT50 values, representing the temperature, at which 50% of plants are killed). Depicted are the markers with significant (p < 0.05) allele effect: BobWhite_c23903_443 (A), BobWhite_c28133_87 (B), Excalibur_c2598_2052 (C), Kukri_c14902_1112 (D), and RAC875_c16644_491 (E). ** indicates significant differences at p < 0.01, *** at p < 0.001, and **** at p < 0.0001. “n” refers to number of observations for each of the two major alleles within the population.”.
Lines 189-192 (Table 1). Table rows were removed after filtering the SNP markers by their effect on phenotypic traits.
Line 195. The sentence “GWAS analyses were conducted to detect significant associations between over-wintering in the field and SNP markers.” was added.
Lines 199-235. The paragraphs were updated to contain relevant information after the additional filtering of SNP markers and investigation of amino acid substitutions.
Line 235. Missing bracket was added to “(Table S6)”.
Line 237. “Figure 5” was changed to “Figure 6”.
Lines 246-248 (Table 2). Table rows were removed after filtering the SNP markers by their effect on phenotypic traits.
Line 271. “(2018)” was removed.
Line 285. “(2012)” was removed.
Line 286. “(2016)” was removed.
Line 302. Space in “Figure 1D” was removed.
Line 307. “(2013)” and “(2015)” were removed.
Line 309. “(2010)” was removed.
Line 311. “(2018)” was removed.
Line 339. The sentences “The SNP results in an amino acid substitution (p.A374V) in this gene. Alanine and valine are both non-polar amino acids, however, valine is more hydrophobic than alanine [61]. The exact effect of these amino acid substitutions should be investigated in further studies.” were added.
Line 352. The sentences “The SNP marker Kukri_c14902_1112 causes a conservative amino acid substitution in this gene (p.I934V). Isoleucine and valine are the most common amino acid substitutions, resulting from a single nucleotide base change [65].” were added.
Line 360. “(2022)’ was removed.
Lines 376-390. Paragraph was deleted.
Line 395. “(2017)” was deleted.
Line 411. “(Figures S6,S7)” was changed to “(Figures S9,S10)”.
Line 434. “12” was changed to “6”.
Line 440. The sentence “Although the SNP is located within the intron of this gene, certain mutations in non-coding regions had previously been shown to lead to disruption of gene translation and RNA splicing, subsequently resulting in an altered protein product [76].” was added.
Line 456. The sentences “The SNP marker BobWhite_c18566_106 leads to an amino acid substitution (p.N2K). Asparagine and lysine are both hydrophilic amino acids. Asparagine to lysine substi-tutions had been shown to affect the conformation of proteins and binding affinity of proteins [81,82].” were added.
Line 516. The sentences “These temperatures were chosen to cover the range from 0% and 100% survival, and thus, to ensure the reliable assessment of LT50 values. This range was established in our earlier studies [34]. Moreover, two Lithuanian winter wheat genotypes (‘Ada’ and ‘Kena DS’) with known LT50 values were included in the freezing tests as a control.” were added.
Lines 546-550. “Figure S6” and “Figure S7” were changed to “Figure S9” and “Figure S10”.
Line 553. “(2019)” was deleted.
Line 556. The sentences “This package provides a number of different GWAS methods. Here, we applied the Bayesian-information and linkage-disequilibrium iteratively nested keyway (BLINK), the fixed and random model circulating probability unification (FarmCPU), the general linear model (GLM), the mixed linear model (MLM), and the multiple loci mixed model (MLMM) [89]” were added.
Line 564. The sentences “Significant SNP markers were additionally filtered by testing the effect of their corresponding two most prevalent alleles on LT50 values and overwintering scores. Markers showing significant effect on a phenotypic trait (p < 0.05) were retained for further analyses. The sequences and synonyms of SNP markers were obtained from the Triticeae Toolbox (T3) repository [90].” were added.
Line 572. The sentences “Amino acid changes were determined by aligning the SNP marker sequences and corresponding gene sequences on MEGA X v. 10.1.7 [93]. MUSCLE alignment algorithm with default parameters was applied.” were added.
Line 584. “Wilcoxon Rank Sum” was included.
Line 586. “(2019)” was deleted; “[43,44]” was included.
Lines 600-604. The significant markers and their locations were updated after an additional filtering step.
Lines 621 – 648. Additional supplementary material descriptions were added.
Line 674. “(2019)” was deleted.
Lines 683-706. Appendix A was updated to contain the relevant marker information after an additional filtering step.
Supplementary material: Table S6. The table was updated to contain two additional columns (“Nucleotide change (position in NCBI Reference sequence)” and “Amino acid change (position in NCBI Reference sequence)”).
Supplementary material: Figures S4,6,8. New supplementary figures were added.
References (lines 708 and onwards). Additional references were included:
Blake, V.C.; Birkett, C.; Matthews, D.E.; Hane, D.L.; Bradbury, P.; Jannink, J. The Triticeae Toolbox: Combining Phenotype and Genotype Data to Advance Small‐Grains Breeding. Plant Genome 2016, 9, doi:10.3835/plantgenome2014.12.0099.
Bricker, J.; Garrick, M.D. An Isoleucine-Valine Substitution in the β Chain of Rabbit Hemoglobin. Biochim. Biophys. Acta - Protein Struct. 1974, 351, 437–441, doi:10.1016/0005-2795(74)90208-6.
Gaffney, D.; Pullinger, C.R.; O’Reilly, D.S.J.; Hoffs, M.S.; Cameron, I.; Vass, J.K.; Kulkarni, M. V.; Kane, J.P.; Schumaker, V.N.; Watts, G.F.; et al. Influence of an Asparagine to Lysine Mutation at Amino Acid 3516 of Apolipoprotein B on Low-Density Lipoprotein Receptor Binding. Clin. Chim. Acta 2002, 321, 113–121, doi:10.1016/S0009-8981(02)00106-7.
Kadowaki, T.; Kadowaki, H.; Accili, D.; Taylor, S.I. Substitution of Lysine for Asparagine at Position 15 in the Alpha-Subunit of the Human Insulin Receptor. A Mutation That Impairs Transport of Receptors to the Cell Surface and Decreases the Affinity of Insulin Binding. J. Biol. Chem. 1990, 265, 19143–19150.
Kumar, S.; Stecher, G.; Li, M.; Knyaz, C.; Tamura, K. MEGA X: Molecular Evolutionary Genetics Analysis across Computing Platforms. Mol. Biol. Evol. 2018, 35, 1547–1549, doi:10.1093/molbev/msy096.
Monera, O.D.; Sereda, T.J.; Zhou, N.E.; Kay, C.M.; Hodges, R.S. Relationship of Sidechain Hydrophobicity and Α‐helical Propensity on the Stability of the Single‐stranded Amphipathic Α‐helix. J. Pept. Sci. 1995, 1, 319–329, doi:10.1002/psc.310010507.
Vaz-Drago, R.; Custódio, N.; Carmo-Fonseca, M. Deep Intronic Mutations and Human Disease. Hum. Genet. 2017, 136, 1093–1111, doi:10.1007/s00439-017-1809-4.

Reviewer 2 Report
Comments and Suggestions for Authors
Explain how did You chose target temperatures (Line 481).
Please describe the metohod You used to genome-wide association studies. The current description is to short.
Author Response
The authors would like to thank the reviewers for their constructive comments and suggestions. We have carefully considered the comments and tried our best to address every one of them. Please find included our responses to the reviewers and the descriptions of any changes made to our manuscript entitled “Genome‐Wide Association Analysis of Freezing Tolerance and Winter Hardiness in Winter Wheat of Nordic Origin”. The line numbers in our responses refer to the lines of the revised manuscript with tracked changes.
Thank you for your consideration of this manuscript.
Response to Reviewer 2
- Explain how did you choose target temperatures (Line 481).
Line 516: The sentences “These temperatures were chosen to cover the range from 0% and 100% survival, and thus, to ensure the reliable assessment of LT50 values. This range was established in our earlier studies [34]. Moreover, two Lithuanian winter wheat genotypes (‘Ada’ and ‘Kena DS’) with known LT50 values were included in the freezing tests as a control.” were included.
- Please describe the method You used to genome-wide association studies. The current description is too short.
Line 556. The sentences “This package provides a number of different GWAS methods. Here, we applied the Bayesian-information and linkage-disequilibrium iteratively nested keyway (BLINK), the fixed and random model circulating probability unification (FarmCPU), the general linear model (GLM), the mixed linear model (MLM), and the multiple loci mixed model (MLMM) [89].” were included.
Line 564. The sentences “Significant SNP markers were additionally filtered by testing the effect of their corresponding two most prevalent alleles on LT50 values and overwintering scores. Markers showing significant effect on a phenotypic trait (p < 0.05) were retained for further analyses. The sequences and synonyms of SNP markers were obtained from the Triticeae Toolbox (T3) repository [90].” were added.
Line 572. The sentences “Amino acid changes were determined by aligning the SNP marker sequences and corresponding gene sequences on MEGA X v. 10.1.7 [93]. MUSCLE alignment algorithm with default parameters was applied.” were added.
Line 584. “Wilcoxon Rank Sum” was included.
List of changes made, by line
(The line numbers refer to the lines of the revised manuscript with tracked changes)
Lines 11-24. The grammar and syntax of the abstract were revised, and the significant markers, as well as their locations, were updated after an additional filtering step.
Lines 13, 252, 293. “winter type” was changed to “winter-type”.
Line 62. “Hetterscheid and Brandenburg in 1995” was changed to “Hetterscheid and Brandenburg [21]”.
Lines 82, 128, 154, 194, 251, 319, 391, 499, 514, 534, 551, 581. All subsection titles were corrected in accordance with the Instructions for Authors.
Line 83. The sentence “To evaluate the FT of 160 Nordic genotypes, freezing tests under controlled conditions were carried out.” was added.
Line 115. “(2019)” was deleted.
Line 117. “Figure 2B” was referenced.
Line 119. “Figure 2 B-D” was changed to “Figure 2 C,D”. Here, we referred to Figure 2 sections “C” and “D”, the figure reference was updated accordingly.
Line 129. The sentence “Field trials were conducted to determine the winter hardiness of the NordGen winter wheat collection.” was added.
Lines 155-171. The sentence “To find significant associations between FT under controlled conditions and SNP markers, GWAS analyses were carried out.” was added. The paragraphs were updated to contain relevant information after the additional filtering of SNP markers and investigation of amino acid substitutions.
Lines 180-187 (Figure 5). new figure was added. Description was included: “Figure 5. The effect of single nucleotide polymorphism (SNP) marker alleles on freezing tolerance (FT) of winter wheat under controlled conditions (LT50 values, representing the temperature, at which 50% of plants are killed). Depicted are the markers with significant (p < 0.05) allele effect: BobWhite_c23903_443 (A), BobWhite_c28133_87 (B), Excalibur_c2598_2052 (C), Kukri_c14902_1112 (D), and RAC875_c16644_491 (E). ** indicates significant differences at p < 0.01, *** at p < 0.001, and **** at p < 0.0001. “n” refers to number of observations for each of the two major alleles within the population.”.
Lines 189-192 (Table 1). Table rows were removed after filtering the SNP markers by their effect on phenotypic traits.
Line 195. The sentence “GWAS analyses were conducted to detect significant associations between over-wintering in the field and SNP markers.” was added.
Lines 199-235. The paragraphs were updated to contain relevant information after the additional filtering of SNP markers and investigation of amino acid substitutions.
Line 235. Missing bracket was added to “(Table S6)”.
Line 237. “Figure 5” was changed to “Figure 6”.
Lines 246-248 (Table 2). Table rows were removed after filtering the SNP markers by their effect on phenotypic traits.
Line 271. “(2018)” was removed.
Line 285. “(2012)” was removed.
Line 286. “(2016)” was removed.
Line 302. Space in “Figure 1D” was removed.
Line 307. “(2013)” and “(2015)” were removed.
Line 309. “(2010)” was removed.
Line 311. “(2018)” was removed.
Line 339. The sentences “The SNP results in an amino acid substitution (p.A374V) in this gene. Alanine and valine are both non-polar amino acids, however, valine is more hydrophobic than alanine [61]. The exact effect of these amino acid substitutions should be investigated in further studies.” were added.
Line 352. The sentences “The SNP marker Kukri_c14902_1112 causes a conservative amino acid substitution in this gene (p.I934V). Isoleucine and valine are the most common amino acid substitutions, resulting from a single nucleotide base change [65].” were added.
Line 360. “(2022)’ was removed.
Lines 376-390. Paragraph was deleted.
Line 395. “(2017)” was deleted.
Line 411. “(Figures S6,S7)” was changed to “(Figures S9,S10)”.
Line 434. “12” was changed to “6”.
Line 440. The sentence “Although the SNP is located within the intron of this gene, certain mutations in non-coding regions had previously been shown to lead to disruption of gene translation and RNA splicing, subsequently resulting in an altered protein product [76].” was added.
Line 456. The sentences “The SNP marker BobWhite_c18566_106 leads to an amino acid substitution (p.N2K). Asparagine and lysine are both hydrophilic amino acids. Asparagine to lysine substi-tutions had been shown to affect the conformation of proteins and binding affinity of proteins [81,82].” were added.
Line 516. The sentences “These temperatures were chosen to cover the range from 0% and 100% survival, and thus, to ensure the reliable assessment of LT50 values. This range was established in our earlier studies [34]. Moreover, two Lithuanian winter wheat genotypes (‘Ada’ and ‘Kena DS’) with known LT50 values were included in the freezing tests as a control.” were added.
Lines 546-550. “Figure S6” and “Figure S7” were changed to “Figure S9” and “Figure S10”.
Line 553. “(2019)” was deleted.
Line 556. The sentences “This package provides a number of different GWAS methods. Here, we applied the Bayesian-information and linkage-disequilibrium iteratively nested keyway (BLINK), the fixed and random model circulating probability unification (FarmCPU), the general linear model (GLM), the mixed linear model (MLM), and the multiple loci mixed model (MLMM) [89]” were added.
Line 564. The sentences “Significant SNP markers were additionally filtered by testing the effect of their corresponding two most prevalent alleles on LT50 values and overwintering scores. Markers showing significant effect on a phenotypic trait (p < 0.05) were retained for further analyses. The sequences and synonyms of SNP markers were obtained from the Triticeae Toolbox (T3) repository [90].” were added.
Line 572. The sentences “Amino acid changes were determined by aligning the SNP marker sequences and corresponding gene sequences on MEGA X v. 10.1.7 [93]. MUSCLE alignment algorithm with default parameters was applied.” were added.
Line 584. “Wilcoxon Rank Sum” was included.
Line 586. “(2019)” was deleted; “[43,44]” was included.
Lines 600-604. The significant markers and their locations were updated after an additional filtering step.
Lines 621 – 648. Additional supplementary material descriptions were added.
Line 674. “(2019)” was deleted.
Lines 683-706. Appendix A was updated to contain the relevant marker information after an additional filtering step.
Supplementary material: Table S6. The table was updated to contain two additional columns (“Nucleotide change (position in NCBI Reference sequence)” and “Amino acid change (position in NCBI Reference sequence)”).
Supplementary material: Figures S4,6,8. New supplementary figures were added.
References (lines 708 and onwards). Additional references were included:
Blake, V.C.; Birkett, C.; Matthews, D.E.; Hane, D.L.; Bradbury, P.; Jannink, J. The Triticeae Toolbox: Combining Phenotype and Genotype Data to Advance Small‐Grains Breeding. Plant Genome 2016, 9, doi:10.3835/plantgenome2014.12.0099.
Bricker, J.; Garrick, M.D. An Isoleucine-Valine Substitution in the β Chain of Rabbit Hemoglobin. Biochim. Biophys. Acta - Protein Struct. 1974, 351, 437–441, doi:10.1016/0005-2795(74)90208-6.
Gaffney, D.; Pullinger, C.R.; O’Reilly, D.S.J.; Hoffs, M.S.; Cameron, I.; Vass, J.K.; Kulkarni, M. V.; Kane, J.P.; Schumaker, V.N.; Watts, G.F.; et al. Influence of an Asparagine to Lysine Mutation at Amino Acid 3516 of Apolipoprotein B on Low-Density Lipoprotein Receptor Binding. Clin. Chim. Acta 2002, 321, 113–121, doi:10.1016/S0009-8981(02)00106-7.
Kadowaki, T.; Kadowaki, H.; Accili, D.; Taylor, S.I. Substitution of Lysine for Asparagine at Position 15 in the Alpha-Subunit of the Human Insulin Receptor. A Mutation That Impairs Transport of Receptors to the Cell Surface and Decreases the Affinity of Insulin Binding. J. Biol. Chem. 1990, 265, 19143–19150.
Kumar, S.; Stecher, G.; Li, M.; Knyaz, C.; Tamura, K. MEGA X: Molecular Evolutionary Genetics Analysis across Computing Platforms. Mol. Biol. Evol. 2018, 35, 1547–1549, doi:10.1093/molbev/msy096.
Monera, O.D.; Sereda, T.J.; Zhou, N.E.; Kay, C.M.; Hodges, R.S. Relationship of Sidechain Hydrophobicity and Α‐helical Propensity on the Stability of the Single‐stranded Amphipathic Α‐helix. J. Pept. Sci. 1995, 1, 319–329, doi:10.1002/psc.310010507.
Vaz-Drago, R.; Custódio, N.; Carmo-Fonseca, M. Deep Intronic Mutations and Human Disease. Hum. Genet. 2017, 136, 1093–1111, doi:10.1007/s00439-017-1809-4.

Reviewer 3 Report
Comments and Suggestions for Authors
Dear authors,
this study is very interesting with scientific importance. The manuscript is about the genome‐wide association analysis of freezing tolerance and winter hardiness in winter wheat of Nordic origin. Freezing tolerance (FT) is evaluated from the panel of 160 winter wheat genotypes of Nordic origin under controlled conditions and 74 of that genotypes from a total of five field trials at two locations in Norway. Genome-wide association studies (GWAS) revealed 6 single nucleotide polymorphism (SNP) markers associated with FT under controlled conditions, and 17 SNP markers associated with field trials. Candidate genes, identified in the study, can be introduced into the breeding programs of winter wheat in the Nordic region.
In the manuscript, introduction and objectives are well and clear written. The materials and methods are given in details and have to be moved from current position on place after the Introduction section. The results obtained and presented in 5 figures and 2 tables are relevant to the proposed objectives. The discussion is appropriate in the context of the results. The conclusions are supported by the results. The references are representative in the field of study.
Before accepting of the manuscript, following parts have to be corrected:
line
4 Gabija Vaitkevičiūtė 1 * > Gabija Vaitkevičiūtė 1,*
82 Subsection titles in the entire manuscript have to be corrected in accordance with the Instructions for Authors.
85 Figure S1 > Figure S1
In the entire manuscript, all cited text of figures and tables in the main text should be without bolding.
107,148,151,167 LT50 > LT50
115 Odilbekov et al. (2019) [22] > Odilbekov et al. [22]
In the entire manuscript, references in the main text should be without year of publishing.
119 Figure 2 B-D > Figure 2 C
262 EEA (2012) > EEA [43]
263 Twardosz and Kossowska-Cezak (2016) > Twardosz and Kossowska-Cezak [44]
279 Figure 2 C,D > Figure 2 C
464 Materials and Method section has to be after the Introduction section
530-532 ?2 and other parts of the formula > no bolding
598 methodology, R.A. A.A., A.C., M.L. and G.V..;
> methodology, R.A., A.A., A.C., M.L. and G.V.;
Author Response
The authors would like to thank the reviewers for their constructive comments and suggestions. We have carefully considered the comments and tried our best to address every one of them. Please find included our responses to the reviewers and the descriptions of any changes made to our manuscript entitled “Genome‐Wide Association Analysis of Freezing Tolerance and Winter Hardiness in Winter Wheat of Nordic Origin”. The line numbers in our responses refer to the lines of the revised manuscript with tracked changes.
Thank you for your consideration of this manuscript.
Response to Reviewer 3
- Line 4 Gabija Vaitkevičiūtė 1 * > Gabija Vaitkevičiūtė 1,*
The comma was included.
- Line 82 Subsection titles in the entire manuscript have to be corrected in accordance with the Instructions for Authors.
All subsection titles were corrected in accordance with the Instructions for Authors.
- Line 85 Figure S1 > Figure S1
The bolding was removed.
- In the entire manuscript, all cited text of figures and tables in the main text should be without bolding.
All citations of figures and tables within the manuscript were changed to regular (unbolded) text.
- Lines 107,148,151,167 LT50 > LT50
The “50” in “LT50” was changed to subscript in the entire manuscript.
- Line 115 Odilbekov et al. (2019) [22] > Odilbekov et al. [22]
The year was removed from the citation.
- In the entire manuscript, references in the main text should be without year of publishing.
The years of publishing were removed from the references in the manuscript, where applicable.
- Line 119 Figure 2 B-D > Figure 2 C
The reference text was unbolded. Here, we referred to Figure 2 sections “C” and “D”, which should be viewed together. The figure reference was updated accordingly.
- Line 262 EEA (2012) > EEA [43]
The year was removed from the reference.
- Line 263 Twardosz and Kossowska-Cezak (2016) > Twardosz and Kossowska-Cezak [44]
The year was removed from the reference.
- Line 279 Figure 2 C,D > Figure 2 C
Here, we referred to Figure 2 sections “C” and “D” both, as one shows the separation according to culton type, and one shows the separation according to level of freezing tolerance. Thus, both sections should be viewed together.
- Line 464 Materials and Method section has to be after the Introduction section
The MDPI “Plants” manuscript template shows that the “Materials and Methods” section should be placed after the “Discussion” section. Our manuscript is in accordance with this requirement.
- Line 530-532 ?2 and other parts of the formula > no bolding
The bolding was removed from the description of the formula.
- Line 598 methodology, R.A. A.A., A.C., M.L. and G.V..; > methodology, R.A., A.A., A.C., M.L. and G.V.;
The comma was included.
List of changes made, by line
(The line numbers refer to the lines of the revised manuscript with tracked changes)
Lines 11-24. The grammar and syntax of the abstract were revised, and the significant markers, as well as their locations, were updated after an additional filtering step.
Lines 13, 252, 293. “winter type” was changed to “winter-type”.
Line 62. “Hetterscheid and Brandenburg in 1995” was changed to “Hetterscheid and Brandenburg [21]”.
Lines 82, 128, 154, 194, 251, 319, 391, 499, 514, 534, 551, 581. All subsection titles were corrected in accordance with the Instructions for Authors.
Line 83. The sentence “To evaluate the FT of 160 Nordic genotypes, freezing tests under controlled conditions were carried out.” was added.
Line 115. “(2019)” was deleted.
Line 117. “Figure 2B” was referenced.
Line 119. “Figure 2 B-D” was changed to “Figure 2 C,D”. Here, we referred to Figure 2 sections “C” and “D”, the figure reference was updated accordingly.
Line 129. The sentence “Field trials were conducted to determine the winter hardiness of the NordGen winter wheat collection.” was added.
Lines 155-171. The sentence “To find significant associations between FT under controlled conditions and SNP markers, GWAS analyses were carried out.” was added. The paragraphs were updated to contain relevant information after the additional filtering of SNP markers and investigation of amino acid substitutions.
Lines 180-187 (Figure 5). new figure was added. Description was included: “Figure 5. The effect of single nucleotide polymorphism (SNP) marker alleles on freezing tolerance (FT) of winter wheat under controlled conditions (LT50 values, representing the temperature, at which 50% of plants are killed). Depicted are the markers with significant (p < 0.05) allele effect: BobWhite_c23903_443 (A), BobWhite_c28133_87 (B), Excalibur_c2598_2052 (C), Kukri_c14902_1112 (D), and RAC875_c16644_491 (E). ** indicates significant differences at p < 0.01, *** at p < 0.001, and **** at p < 0.0001. “n” refers to number of observations for each of the two major alleles within the population.”.
Lines 189-192 (Table 1). Table rows were removed after filtering the SNP markers by their effect on phenotypic traits.
Line 195. The sentence “GWAS analyses were conducted to detect significant associations between over-wintering in the field and SNP markers.” was added.
Lines 199-235. The paragraphs were updated to contain relevant information after the additional filtering of SNP markers and investigation of amino acid substitutions.
Line 235. Missing bracket was added to “(Table S6)”.
Line 237. “Figure 5” was changed to “Figure 6”.
Lines 246-248 (Table 2). Table rows were removed after filtering the SNP markers by their effect on phenotypic traits.
Line 271. “(2018)” was removed.
Line 285. “(2012)” was removed.
Line 286. “(2016)” was removed.
Line 302. Space in “Figure 1D” was removed.
Line 307. “(2013)” and “(2015)” were removed.
Line 309. “(2010)” was removed.
Line 311. “(2018)” was removed.
Line 339. The sentences “The SNP results in an amino acid substitution (p.A374V) in this gene. Alanine and valine are both non-polar amino acids, however, valine is more hydrophobic than alanine [61]. The exact effect of these amino acid substitutions should be investigated in further studies.” were added.
Line 352. The sentences “The SNP marker Kukri_c14902_1112 causes a conservative amino acid substitution in this gene (p.I934V). Isoleucine and valine are the most common amino acid substitutions, resulting from a single nucleotide base change [65].” were added.
Line 360. “(2022)’ was removed.
Lines 376-390. Paragraph was deleted.
Line 395. “(2017)” was deleted.
Line 411. “(Figures S6,S7)” was changed to “(Figures S9,S10)”.
Line 434. “12” was changed to “6”.
Line 440. The sentence “Although the SNP is located within the intron of this gene, certain mutations in non-coding regions had previously been shown to lead to disruption of gene translation and RNA splicing, subsequently resulting in an altered protein product [76].” was added.
Line 456. The sentences “The SNP marker BobWhite_c18566_106 leads to an amino acid substitution (p.N2K). Asparagine and lysine are both hydrophilic amino acids. Asparagine to lysine substi-tutions had been shown to affect the conformation of proteins and binding affinity of proteins [81,82].” were added.
Line 516. The sentences “These temperatures were chosen to cover the range from 0% and 100% survival, and thus, to ensure the reliable assessment of LT50 values. This range was established in our earlier studies [34]. Moreover, two Lithuanian winter wheat genotypes (‘Ada’ and ‘Kena DS’) with known LT50 values were included in the freezing tests as a control.” were added.
Lines 546-550. “Figure S6” and “Figure S7” were changed to “Figure S9” and “Figure S10”.
Line 553. “(2019)” was deleted.
Line 556. The sentences “This package provides a number of different GWAS methods. Here, we applied the Bayesian-information and linkage-disequilibrium iteratively nested keyway (BLINK), the fixed and random model circulating probability unification (FarmCPU), the general linear model (GLM), the mixed linear model (MLM), and the multiple loci mixed model (MLMM) [89]” were added.
Line 564. The sentences “Significant SNP markers were additionally filtered by testing the effect of their corresponding two most prevalent alleles on LT50 values and overwintering scores. Markers showing significant effect on a phenotypic trait (p < 0.05) were retained for further analyses. The sequences and synonyms of SNP markers were obtained from the Triticeae Toolbox (T3) repository [90].” were added.
Line 572. The sentences “Amino acid changes were determined by aligning the SNP marker sequences and corresponding gene sequences on MEGA X v. 10.1.7 [93]. MUSCLE alignment algorithm with default parameters was applied.” were added.
Line 584. “Wilcoxon Rank Sum” was included.
Line 586. “(2019)” was deleted; “[43,44]” was included.
Lines 600-604. The significant markers and their locations were updated after an additional filtering step.
Lines 621 – 648. Additional supplementary material descriptions were added.
Line 674. “(2019)” was deleted.
Lines 683-706. Appendix A was updated to contain the relevant marker information after an additional filtering step.
Supplementary material: Table S6. The table was updated to contain two additional columns (“Nucleotide change (position in NCBI Reference sequence)” and “Amino acid change (position in NCBI Reference sequence)”).
Supplementary material: Figures S4,6,8. New supplementary figures were added.
References (lines 708 and onwards). Additional references were included:
Blake, V.C.; Birkett, C.; Matthews, D.E.; Hane, D.L.; Bradbury, P.; Jannink, J. The Triticeae Toolbox: Combining Phenotype and Genotype Data to Advance Small‐Grains Breeding. Plant Genome 2016, 9, doi:10.3835/plantgenome2014.12.0099.
Bricker, J.; Garrick, M.D. An Isoleucine-Valine Substitution in the β Chain of Rabbit Hemoglobin. Biochim. Biophys. Acta - Protein Struct. 1974, 351, 437–441, doi:10.1016/0005-2795(74)90208-6.
Gaffney, D.; Pullinger, C.R.; O’Reilly, D.S.J.; Hoffs, M.S.; Cameron, I.; Vass, J.K.; Kulkarni, M. V.; Kane, J.P.; Schumaker, V.N.; Watts, G.F.; et al. Influence of an Asparagine to Lysine Mutation at Amino Acid 3516 of Apolipoprotein B on Low-Density Lipoprotein Receptor Binding. Clin. Chim. Acta 2002, 321, 113–121, doi:10.1016/S0009-8981(02)00106-7.
Kadowaki, T.; Kadowaki, H.; Accili, D.; Taylor, S.I. Substitution of Lysine for Asparagine at Position 15 in the Alpha-Subunit of the Human Insulin Receptor. A Mutation That Impairs Transport of Receptors to the Cell Surface and Decreases the Affinity of Insulin Binding. J. Biol. Chem. 1990, 265, 19143–19150.
Kumar, S.; Stecher, G.; Li, M.; Knyaz, C.; Tamura, K. MEGA X: Molecular Evolutionary Genetics Analysis across Computing Platforms. Mol. Biol. Evol. 2018, 35, 1547–1549, doi:10.1093/molbev/msy096.
Monera, O.D.; Sereda, T.J.; Zhou, N.E.; Kay, C.M.; Hodges, R.S. Relationship of Sidechain Hydrophobicity and Α‐helical Propensity on the Stability of the Single‐stranded Amphipathic Α‐helix. J. Pept. Sci. 1995, 1, 319–329, doi:10.1002/psc.310010507.
Vaz-Drago, R.; Custódio, N.; Carmo-Fonseca, M. Deep Intronic Mutations and Human Disease. Hum. Genet. 2017, 136, 1093–1111, doi:10.1007/s00439-017-1809-4.

Round 2
Reviewer 1 Report
Comments and Suggestions for Authors
All my questions have been carefully addressed by the author.